# A Collaborative Platform for Advancing Automatic Interpretation in ECG Signals

**DOI:** 10.3390/diagnostics14060600

**Published:** 2024-03-12

**Authors:** Luis Alberto Gordillo-Roblero, Jorge Alberto Soto-Cajiga, Daniela Díaz-Alonso, Francisco David Pérez-Reynoso, Hugo Jiménez-Hernández

**Affiliations:** 1Center for Engineering and Industrial Development (CIDESI), National Laboratory for Research on Medical Technologies (LANITEM), Pie de la Cuesta 702, Des. San Pablo, Querétaro 76125, Mexico; jsoto@cidesi.edu.mx (J.A.S.-C.); daniela.diaz@cidesi.edu.mx (D.D.-A.); investigador3.lanitem@cidesi.edu.mx (F.D.P.-R.); 2Faculty of Informatics, Autonomous University of Querétaro (UAQ), Cerro de las Campanas S/N, Querétaro 76010, Mexico; hugo.jimenez@uaq.mx

**Keywords:** electrocardiography, ECG, automatic interpretation, signal processing, open hardware, electrocardiograph, collaborative platform

## Abstract

Numerous papers report the efficiency of the automatic interpretation capabilities of commercial algorithms. Unfortunately, these algorithms are proprietary, and academia has no means of directly contributing to these results. In fact, nothing at the same stage of development exists in academia. Despite the extensive research in ECG signal processing, from signal conditioning to expert systems, a cohesive single application for clinical use is not ready yet. This is due to a serious lack of coordination in the academic efforts, which involve not only algorithms for signal processing, but also the signal acquisition equipment itself. For instance, the different sampling rates and the different noise levels frequently found in the available signal databases can cause severe incompatibility problems when the integration of different algorithms is desired. Therefore, this work aims to solve this incompatibility problem by providing the academic community with a diagnostic-grade electrocardiograph. The intention is to create a new standardized ECG signals database in order to address the automatic interpretation problem and create an electrocardiography system that can fully assist clinical practitioners, as the proprietary systems do. Achieving this objective is expected through an open and coordinated collaboration platform for which a webpage has already been created.

## 1. Background

Electrocardiography has been a very active research topic practically since the invention of the electrocardiograph. Research on this topic has been equally intense in both physiology and in the signal processing field. Physiologists’ concern has been mostly to determine how to interpret the electrical activity of the heart using the electrocardiogram (ECG), while other scientists’ objective has been the continuous improvement of equipment and algorithms for signal processing. These two broad areas of research have formed a prolific relationship, where advancements on one side usually lead to advancements in the other.

For the purpose of this proposal, it is very important to mention that the number of pathologies and other clinical parameters that can be detected and measured using the electrocardiogram is huge. In fact, many books on the interpretation of the electrocardiogram have been written. These books are usually written in a clear and condensed way [1,2] that makes the characteristics of pathologies easy to understand, and consequently, easy to translate into computer logic. Unfortunately, due to the number of pathologies and how different clinical parameters are related to each other, even brilliant minds can find it hard to remember and detect all possible abnormalities while exploring an electrocardiogram. For this reason, computer algorithms to assist human physiologists in the ECG interpretation process are now a need for the sake of both patients and clinical practitioners.

Signal processing is by itself a very active area of research. Publications in this area are related to hardware development to some extent, but the biggest part involves the development of algorithms for solving a wide spectrum of different signal processing problems, ranging from signal conditioning to expert systems. Given the thousands of published papers presenting algorithms for ECG signal processing, one may think that everything in this field has been conducted already. But if the evaluation of these algorithms in real clinical applications is desired, one finds that most of the algorithms are not even ready for clinical use as they were designed for solving very specific problems. However, in clinical practice, much more than solving a specific problem is required. Therefore, the goal of this proposal is to leverage efficiency in the automatic interpretation of the electrocardiogram but in a way that can be useful in the clinical practice, as this is where the progress and the real efficiencies of the many developed algorithms can be condensed and evaluated. In order to reach this goal, much is yet to be done.

The most advanced algorithms for automatic interpretation available in the market are the GE Marquette™ 12SL™ [3,4] and the DXL ECG ™ algorithm [5], which belong to General Electric and Philips, respectively. These algorithms are proprietary; they were designed for clinical use, and in academia, nothing at the same level of development exists. Even though many algorithms for solving several different problems have been published in different academic journals, these algorithms are not yet integrated into a single machine, and therefore they cannot be used in clinical practice.

Developing an algorithm like the two aforementioned is a titanic work that requires coordination, integration, and a certain degree of standardization. These tasks have been performed publicly by a well-known project called PhysioNet [6], which is currently the worldwide reference for ECG signal processing research. However, in its repository of challenges, there is not yet an explicit intention of developing an algorithm to compete with the most advanced commercial algorithms available in the market. This is the intention of this proposal, “The Kenshin Project”.

The Kenshin Project offers, as a starting point, an open hardware platform whose main purpose is the standardization of an ECG signal databank. Standardized signals can ease the integration of different algorithms, avoiding the unnecessary work of adapting algorithms to different signal characteristics, and enabling them to run under the same computing environment. The possibility of having standardized algorithms (application code) in addition to some project coordination will also avoid having to work twice on the same specific problem once a high level of efficiency has been reached. This will allow academia’s future efforts to focus on advancing the stage of automatic interpretation. Furthermore, the hardware proposed here will also enable the transfer of academia’s results to regular clinical practitioners.

## 2. Problem Definition

Multiple publications report a variety of efficiencies measured for the automatic interpretation capabilities of commercial electrocardiographs, to cite a few of many: [7,8,9,10,11,12]. In [11], one can see that for the case of the sinus rhythm detection, which is the most common use of the electrocardiogram, a study was conducted by randomly recording 2112 electrocardiograms in order to later run a computer-based interpretation process on them, using one of the most prominent commercial systems [3,4]. The overall efficiency reported for this specific case (sinus rhythm detection) was 88%. The efficiency for ventricular rhythms was 95%, and the efficiency for non-ventricular rhythms was reported at less than 54%. Although the reported 95% efficiency seems to be good, unfortunately, in clinical practice, there is no way to ensure that only ventricular rhythms will be analyzed. In this sense, the more conservative 88% number is not yet reliable, and less than 54% for non-ventricular rhythms is completely unacceptable. It should be noted that the last cited papers evaluated commercial ECG interpretation algorithms. This is important because these algorithms are not disclosed, and currently, there is no way for the academia to directly contribute to these results.

In [10], the most popular computer-aided ECG analysis problems were reviewed, focusing attention on real-world clinical applications. As a conclusion in this paper, the proposal of new evaluation and analysis methodologies is suggested for new research, and it is unsurprising that even business-related issues were also mentioned there. In [7], seven different ECG-analysis programs were tested on more than 2000 ECGs. In this study, if any of the seven programs produced different analysis results, the ECG would then be reviewed by a group of expert cardiologists. Different conditions like rhythm classification, identification of atrial fibrillation/flutter, and the ability to flag an ECG for further review were evaluated. The study showed that the agreement between programs and the majority reviewer’s decision ranged from 46% to 62%. Therefore, from this result, in [7] an explicit call for not relying on automatic interpretation procedures alone was made to healthcare institutions. Other papers, like [8], report the low efficiencies and their clinical consequences for the specific case of atrial fibrillation, and papers, like [9], call for a closer collaboration between clinical experts and manufacturers. The list can go on and on.

Until now, the problem to be solved with the help of this proposal has been partially stated in the previously cited papers: low efficiency in the clinical practice. In addition to this, and as a part of the same problem, it is important to consider that the number of pathologies and other useful clinical parameters described in the books about electrocardiogram interpretation can be counted by the dozens [2], and the vast majority have not yet been considered in the entire repository of academic publications regarding ECG signal processing. For example, it is not possible to find an algorithm or method for automatically detecting a sick sinus syndrome (p. 78 of [1]) or a first-degree Atrio-Ventricular Block (p. 155 of [1]) using a scholarly search engine. Awareness of this situation will help to disregard the common tendency of believing that in the electrocardiogram only one waveform exists. This would be useful since, in order to detect the many different available pathologies, it is necessary to be able to recognize different waveforms first [13]. These waveforms are often found in different ECG-leads like a Bundle-Branch Block (p. 264 of [1]), and sometimes even require comparing waveforms for 3D-locating parts of the heart.

Generally, commercial equipment is proprietary, and neither its hardware nor its software is disclosed to academic researchers. Philips and General Electric have their own research centers [14,15], and their commercial ECG algorithms [4,5] are protected by several patents [16,17]. It is not possible to know to what degree they use academic publications to advance their algorithms or if these private research centers have ever published their complete research. Therefore, all the efficiency evaluations performed on commercial equipment are meant to be only for informational purposes, as academia has no means to contribute to proprietary research. That is why the main objective of this project’s proposal is to collectively (and publicly) create an algorithm capable of emulating a human expert. This will provide access to this applied research field and will help to discourage the current tendency of solving very specific problems with solutions that are later difficult to share and integrate with other procedures. Fortunately, commercial equipment has been the subject of multiple studies reporting efficiency in clinical practice, as this will serve as our project’s reference point.

It is well known that research can go in many directions in exploring the unknown. It is also understood that the specificity of a problem often enhances the quality of the research. However, achieving high efficiency in the automatic interpretation of the ECG at the clinical level is a significant challenge, and if its development is left to chance, this objective probably will never be reached. Keeping the general problem in mind will help in developing methodologies that can be effectively used across a wider range of different problems [10], avoiding method incompatibility issues and serving as a guide for optimizing efforts. For example, sixty years have passed since the oldest article available in the IEEE Xplore Library was published regarding ECG automatic interpretation [13], and despite the time and the intense research, the problem addressed in this aforementioned paper [13] is not yet solved, which is the need to recognize different waveforms in order to extract their physiological parameters.

The current state of the art in ECG automatic interpretation is clearly the result of a significant lack of coordination in academia’s efforts. Automatic ECG interpretation is a complex problem that cannot be treated as a single, specific problem. It requires the integration of many specific problem solutions, which need to be compatible in order to work together. Therefore, it can be said that solving a problem like this requires coordination, planning, integration, and standardization. In fact, a plea for standardization was made a long time ago [18], a plea that has been overlooked by commercial medical equipment manufacturers worldwide, perhaps due to business-related concerns. For the Kenshin Project, it is of utmost importance to keep the pursued general objective clearly in mind, as this will dictate the guidelines, requirements, and specific objectives to facilitate collaboration. This will expedite the process of developing a system that can interpret the electrocardiogram as efficiently as a human expert would do. The project’s intention is to fully assist human practitioners, not to explicitly provide a diagnosis, in order to avoid ethical or legal issues [19].

At the current starting stage of the project, it was decided to begin solving the biggest hardware-related problem, which is the standardization of the signals. Although PhysioNet [6] has a vast bank of signals made available through the collaboration of many researchers, the signals are not standardized. Some signals are being sampled at 100 Hz [20], others at 250 Hz [21], others at 500 Hz [22], others at 1 kHz [23], and so on [24]. Some signals are annotated by human experts [21], and some others are not [23]. Some signals have accepted levels of system noise [23] and others are very noisy [21]. Moreover, the existing convention for annotating signals proposed in PhysioNet presents several gaps [10]. In summary, sampling rates should depend on the signal’s nature. ECG signals have a thin, but time-varying, bandwidth [25]. The AHA (American Heart Association) recommended bandwidth for standard clinical applications is in the range of 0.05–125 Hz [26], up to 250 Hz for pediatrics [27] and up to 1 kHz for pacemaker detection [27,28]. These AHA recommendations have proven their accuracy even in modern publications where the sampling-rate plays an important role in the analysis of time-critical pathologies such as Heart-Rate Variability [29].

A good design recommendation would be to keep the sampling rate low enough to avoid high-frequency noises from entering the system, but without affecting the physiological accuracy. From information given and according to the Nyquist theorem, sampling rates below the edge of 300 Hz for a standard ECG (or 500 Hz for pediatrics) are not reliable for automatic interpretation procedures. However, some authors decided to use the available signals sampled at 250 Hz [21] in their proposals, which is not appropriate for diagnostic applications. Good signals should comply with certain noise levels stated in different American Standards [27,30,31,32]. In fact, if the acquisition system uses the Right-Leg Lead to improve the Common-Mode Rejection Ratio as described in Section A.4.2.9.3 of [31] and on page 1116 of [27], the system noise can be maintained as low as 30 µV and should not be greater than that, as it is specified in Section 4.2.9.3 of the ANSI/AAMI EC13:2002 Standard [31].

Standardization of signals can help to integrate algorithms, thereby completely eliminating the sampling rate incompatibility problem. Even though algorithms to homogenize the sampling rate could be implemented to remediate incompatibility during the research process, doing so is unnecessary work and can cause more problems than the one it solves. For example, for an online application where the output of a filter can be the input for a waveform recognition process, having a sampling rate adapter in between these two processes delays the signal and increases computer consumption, which will cause the algorithm to run slower. For offline applications, the use of an algorithm to match sampling rates can be used without major problems, but it is still unnecessary. An example of an offline application could be when a signal is acquired with an electrocardiograph and the interpretation process is executed in a different system.

Different system-noise levels in ECG signals can also be a problem, perhaps even worse than the sampling rate incompatibility, mainly because the performance of most filters depends on some design parameters that are adjusted depending on the signal noise. Feeding a filter with signals that have significantly different system noise levels will result in a noisier output for any of them for which the filter was not adjusted. This will cause a problem for a subsequent process. For example, a designed waveform recognition algorithm could work fine with a certain level of system noise, but might not work at all if signals are noisier than expected. Not to mention, the fact that algorithms for automatically adjusting filter parameters could even be more complicated than those designed for simply matching a sampling-rate. All this, again, will delay the signals and increase computing consumption.

It is known that ECG signals contain subtle information embedded in very specific parts of their waveforms, which somehow should be preserved. Most filters can cause a waveform deformation on signals, especially those that are frequency-based. If signals are very noisy, in an attempt to make the signal look smoother, designers tend to use a more aggressive method for filtering that could cause a greater signal distortion. This deformation can affect the accuracy in measuring parameters of pathologies like the ST-segment elevation [33] or Heart Rate Variability Analysis [34] where the slightest variation in time can be an indication of a severe heart disease. More dramatically, the QT-segment prolongation, which is also a time-based parameter, is a powerful mortality marker [35]. Due to these important clinical conditions, not knowing how signals were processed before making them available for the researcher can severely impact the accuracy of physiological research. Therefore, having the raw ECG signal for further physiological reference will be very useful. It is worth mentioning that even when non-frequency-based algorithms have been proposed with the intention of preserving the physiological characteristics of the signals [36], these algorithms still need to use a time buffer to adjust their smoothness capability, which ultimately depends on system noise levels.

The possibility of encountering the aforementioned situations (sampling rate differences and different system noise levels) while conducting research using the PhysioNet signal databank is extremely high. Dealing with both of these situations at the same time can indeed worsen the signal processing panorama, sometimes making impossible to integrate two different sequential processes. Therefore, in order to avoid these situations, having a databank with standardized sampling rate, in addition to having ANSI/AAMI compliant noise levels, would be beneficial for integration and collaboration purposes. To coordinate efforts, a webpage for the Kenshin Project has been created (https://kenshin-ai.org).

## 3. Solution

The focus of this proposal is on addressing the signal standardization problem. Both of the previously-mentioned problems, namely the sampling rate and the system-noise levels, can be solved at the same time since they are both entirely dependent on the device hardware. However, a more comprehensive problem description has been provided in the preceding section, with the intention of outlining the general approach in which the Kenshin Project is planning to provide the community with a reliable electrocardiography system. This is crucial because it helps to lay the foundations for future work.

The proposed solution to the signal’s incompatibility problem involved the design and manufacture of an electrocardiograph device in order to make it available to the academic community. This new device will enable the acquisition of new ECG signals for integration into a standardized signals databank. The device’s name is Kenshin, its design and manufacture are complete (see Figure 1 and Figure 2), and its full documentation has been published at https://kenshin-ai.org where it is already available for community use and academic collaboration.

The just developed electrocardiograph is based on a Front-End Solution offered by Analog Devices, Inc. (Wilmington, MA, USA) [37]. It has a USB 2.0 type C interface based on a 32-bit Floating Point Microcontroller from Microchip Technologies, Inc. (Chandler, AZ, USA) [38]. It can be powered with an external medical grade power supply, but it can also be powered from the USB port. It is compact and simple in design, and it was carefully designed to meet EMC requirements, as well as to comply with a variety of different standards. A device connectivity diagram is shown in Figure 3.

As can be seen in Figure 3, this device was designed to be affordable. The cost of having an embedded operating system (OS) with all the necessary hardware has been eliminated by the use of a computer or tablet as the main signal processor. Some of the eliminated devices include an embedded display, input devices, a larger power supply, and greater processor capacity. In fact, modern commercial computing devices have higher screen resolutions than what can be feasibly embedded in an ECG machine at a similar cost. Many computers also have the capability of driving multiple displays. Medical grade input devices, like keyboards and mice, are available on the market. Medical grade accessories like carts, cabinets, and trolleys are also available, allowing this electrocardiograph to be configured as a complete diagnostic unit. It is also important to mention that since ECG signals have an even lower bandwidth than audio signals, it is expected that commercial processors can handle all the required computing without any problem.

In addition to the low-cost advantage, the lack of attachment to a specific operating system, coupled with the versatility of the USB port, makes this newly-designed electrocardiograph capable of operating with any operating system, such as Android, Windows, or Linux. This opens the possibility to develop applications that leverage the unique advantages of each OS. For instance, Android would be suitable for ambulatory applications, while Windows could be ideal for diagnostic applications. However, it is important to note that researchers in the physiological field may not be experts in managing computer applications. Therefore, it’s recommended to design user-friendly interfaces in user-friendly operating systems, even for research purposes.

The core of this electrocardiograph is a set of two ADAS1000 Analog Front-End Integrated circuits [37]. These circuits are configured in a master/slave configuration to achieve a standard 12-lead electrocardiograph. According to the manufacturer’s datasheet, the ADAS1000 ICs can support the standards AAMI EC11 [30], AAMI EC38 [39], AAMI EC13 [31], IEC60601-1 [40], IEC60601-2-25 [41], IEC60601-2-27 [42], and IEC60601-2-51 [43]. This ensures that the newly-designed electrocardiograph is a diagnostic-grade device. The parameters considered in choosing the Analog Devices’ solution over other Front-End options include: the more reliable CMMR, which is 110 dB (typical) measured with an AAMI/IEC standard, the lower system noise at its best performance, which is 6 µV, and a couple more supported standards. The comparison was based on the values reported in their respective datasheets. Besides these previously mentioned parameters, all major front-end solutions providers offer practically the same features.

When designing an electrocardiograph, it is difficult to achieve the same performance offered by Front-End solutions using independent instrumentation amplifiers while maintaining the same level of integration, performance, and cost. To ensure that the ADAS1000 is ready for use in clinical applications, additional auxiliary circuits were added. These circuits include defibrillator protection, hardware-based analog filters, the Right-Leg Drive calibration feedback and electrostatic discharge protection for all exposed parts. It is important to note that the ADAS1000 includes a shield-driver and lead-off detection circuits, which were made ready-to-use in the Kenshin Device. Furthermore, in order to achieve the highest signal quality, design considerations were meticulously taken into account, both for the electrical design and the quality of components, as well as to meet applicable EMC requirements. A functional block diagram of the Kenshin Device is shown in Figure 4.

The ECG-cable set was designed without series resistors because the defibrillator protection was implemented on-board using a specialty IC offered by Maxim Integrated [44]. The cable color code was chosen to be AHA compatible and, since there is no standard for connectors, the most conventional DA-15 pinout configuration was used. Low pass filters in all analog channels were designed to have an upper cut-off frequency of 21 kHz to allow pacemaker detection. Low ESR (Equivalent Series Resistance) X2Y capacitors were chosen for this filter, and their ground plates were connected to a solid, quiet ground away from digital circuits. All channels were routed in matched-length mode, from the electrodes to the low-pass filters and then from the filters to the ADAS1000 analog inputs. This was performed to maintain the CMRR (Commom Mode Rejection Ratio) as high as possible.

Communication between ADAS1000 ICs and the microcontroller is conducted using three independent SPI interfaces. The master ADAS1000 has two SPI interfaces: one is used for diagnostic signals, and the other, with a higher sampling rate, is used for pacemaker detection. The slave device has only one SPI interface. PCB traces for these interfaces were designed in a controlled impedance manner using series termination resistors to avoid reflections, thereby preventing EMC (Electro-Magnetic Compatibility) issues. The preferred sampling rate for diagnostic signals in the Kenshin Device is 2 kHz because this configuration offers the lowest system noise. The sampling rate for pacemaker detection is fixed at 128 kHz. In both cases, the standard’s required bandwidth is fulfilled. The ADAS1000 circuits have three predefined sampling rates for diagnostic signals (2 kHz, 16 kHz, and 128 kHz), presenting a trade-off between power consumption and signal quality. For our intended automatic interpretation purposes, signal quality was given priority over power consumption.

The microcontroller used is the Microchip PIC32MZEF2040 [38]. Its main task is to provide the electrocardiograph with native USB 2.0 High-Speed connectivity and to serve as the link between the microcontroller and the ADAS1000 ICs through the SPI interfaces. It has a maximum core frequency of 200 MHz. Additionally, if needed, the microcontroller is equipped with a Floating Point Unit for signal processing operations. It also has 512 KB of RAM and 2 MB of Flash Memory. This microcontroller was chosen due to its low cost and the simplicity of its programming interface. It can be programmed in-circuit using an RJ11 connector, and its development environment is provided free of charge by the manufacturer.

The power supply of the Kenshin Device is somewhat complex due to the USB-C specification. Multiple voltage monitors and a single current limiter are used to initiate power to the device from the USB port. A power multiplexer is included in case the device needs to be powered from a medical-grade external source. The switch from USB power to an external power supply is automated, with preference given to the external source. A medical-grade power supply can enable the Kenshin Device to meet the requirements defined by the standard EN/IEC/UL60601-1, which stipulates that isolation levels need to be at 2-MOPP (Means of Patient Protection).

The power supply’s initial entry is a buck booster. It was included to stabilize voltage level variations from the USB port or an external power supply, providing linear voltage regulators with a steady input that can be further improved by the regulators themselves. Special care was taken in selecting these linear voltage regulators. The regulator used to power analog circuits is the LP5907 [45], which was specifically designed for analog or RF applications. Each analog IC has its own linear regulator. The regulator for powering digital circuits is the NCV8161 [46], which has a greater current capability and still maintains a good PSRR (Power Supply Rejection Ratio). To comply with the USB specification, which states that no more than 100 mA can be drawn from the port during device enumeration, the digital part is powered first. Once device enumeration is completed, 500 mA is requested from the USB port. After this, the analog linear regulators can be enabled using two independent microcontroller output pins. These pins can also be used to power down the ADAS1000 integrated circuits when needed.

The PCB layout was carefully planned in order to avoid EMC issues. Four layers were considered sufficient to meet the design requirements. Layer two was chosen to be a solid ground plane along the entire board. This plane has two main sections: the digital section, where the microcontroller and power supply are located, and the analog section, where only the ECG cable and its corresponding circuitry are connected. On the digital side, auxiliary ground planes were placed on different layers. For the power supply section, the auxiliary ground plane was placed in layer 3. In this way, power and signal traces for the power supply were routed only in layers 1 and 4. For the digital signals section, the auxiliary ground plane was placed in layer 4. In this way, digital traces were routed only on layers 1 and 3, and the auxiliary ground plane in layer 4 served as a ground shield that allowed the routing of SPI interfaces in strip-line configuration with a controlled impedance of 50 ohms. Series termination resistors were added to high-speed digital inputs, and some resistors were added to match impedance between IC pins and traces, as done in the USB differential pair. The overall PCB thickness is 0.6 mm and the distance between layers (layer stackup) is shown in the fabrication drawing published on the project’s webpage. PCB thickness was chosen to meet the controlled impedance requirement and also to follow the design guidelines for USB layout given by Texas Instruments in application note 26.2 [47]. Shielding fences were added around the buck-booster circuit to reduce radiated emission. Shield fences were added around the perimeter of the entire board to reduce received emissions, and a third fence was used to provide shielding between the digital and analog parts of the board. A plane clearance was added in all layers under the power inductor used by the buck booster to prevent its radiation from polluting the ground planes. On the analog side of the board, layers 2 and 3 are ground planes; they were defined in this way to provide a solid plane on which to run analog signals. An auxiliary shielding board was placed on the bottom side of the main printed circuit board with the intention of reducing ESD interference in objects near the electrocardiograph.

## 4. Results

The newly designed electrocardiograph was first tested with the test tones generated by the ADAS1000 itself. To perform this test, the ECG channel configuration was set to be “electrode mode”, as described in the datasheet [37]. The test tone signals were routed internally to the non-inverting input of each differential amplifier and the inverting inputs of all available channels were routed to the VCM_REF voltage, which is a fixed 1.3 V internal voltage reference. Ten different channels are available in this configuration. Five correspond to the master device (ADAS1000) and another five correspond to the slave device (ADAS1000-2). The test tone is generated by the Master device and it is shared with the slave using a copper trace to a corresponding input. All three possible waveforms were tested. The results are shown in Figure 5, Figure 6, Figure 7, Figure 8. All filters were disabled.

The purpose of presenting the test tones is to demonstrate the system noise and its effects at different frequencies. In the square wave (Figure 5), the system noise, commonly known as “grass noise”, is clearly visible. The procedure for calculating the system noise using this waveform first involves calculating an average of the high level of the square wave, as well as an average for the low level. The difference between these two calculated averages should be 1mV, as specified in the datasheet. Dividing 1mV by the number of elements between the two averaged values (9.909016032 × 10^3^) provides a measurement unit, which is the system resolution. The calculated resolution was 0.100918193 µV/LSB (Least Significant Bit).

Next, the average for all the noise peaks was calculated, as was the average for all the noise valleys. Finally, the difference between these last calculated averages, multiplied by the system resolution, provides an estimation of the system noise. This methodology is illustrated in Figure 9. For the channels contained in the Master ADAS1000, it was in average 5 µV p-p, and for the channels contained in the slave device, it was also 5 µV p-p, which is the same, even though for the slave device the test tone had to travel over a copper trace. It is important to note that in both cases, the noise is kept under 30 µV as stated in the standard [31]. From Figure 5, it can be deduced that the power supply and ground distribution are clean and reliable for clinical use. This performance was achieved due to the considerations taken during the Printed Circuit Board design.

For the case of the sine wave, 1 mV p-p, 10 Hz (Figure 6), it can be seen that the system noise is somewhat hidden by the speed of the signal and the sampling rate, specially where the signal is slow (the peaks or valleys). For this case, it is more difficult to numerically estimate a level of system noise, but from a visual inspection, it can be observed that it is still the same (5 µV) for both ICs. In the case of the other sine wave, 1mV p-p, 150 Hz (Figure 7), it can be seen that the system noise is totally masked by the signal speed and the sampling rate. It is important to note that for the last three cases, it is not possible to calculate a signal-to-noise ratio because no signal generator or measuring instrument is completely clean or precise, nor is it possible to manually superimpose an ideal signal over the acquired signal, because doing this will cause an error by itself. The best measuring reference available is what is specified in the device datasheet [37], and the best place for estimating the system noise was the square wave test tone. In the design documentation published on the project’s webpage, MATLAB (R2022a, 9.12.0) workspaces with all the acquired data are provided.

For testing the device with ECG signals, the FLUKE PS420 signal simulator was used. The purpose of using a signal simulator is, again, to evaluate the device’s system noise using clean signals. Needless to say, the simulator provides the cleanest signals one can find. The cable used was a DA-15 ECG cable without resistors for defibrillator protection. This was performed like this because a very precise defibrillator protection circuit was included in the Kenshin Device. The configuration used for testing the device with ECG signals was also described in the datasheet as “electrode mode”. This mode was chosen because in this configuration, we can see the performance of each differential amplifier with no other processing than the differential amplifier itself. The non-inverting input of each differential amplifier is connected to the signal, and the inverting input was connected to the WCT (Wilson Center Terminal). The contributing electrodes for the RLD amplifier were chosen to be the same as for the WCT (Right-Arm, Left-Arm, and Left-Leg). Finally, with the intention of evaluating the hardware performance, all filters were disabled. Results are shown in Figure 8.

From Figure 8, it can be noticed that the channels corresponding to RA, LA, LL, and V1 are slightly noisier than expected; the estimated system noise for these channels is 30 µV. These channels correspond to the first four channels of the master device. For the rest of the channels, including those in the slave device, the estimated system noise is again 5 µV, the same as the estimated system noise in the square wave test tone. The reason for this difference could be an issue with the PCB design; the noisy channels have a single via along the channel, while the rest of the channels do not. All channels were length-matched and routed identically, except for the additional via. For further reference, the fabrication files are available at https://kenshin-ai.org. And of course, this issue will be corrected in the next Kenshin Device model. Later, another cable with 10k resistors for defibrillation protection was used with no appreciable differences. In fact, another FLUKE ECG signal simulator, the ProSim 4, was also used with no remarkable differences compared to the ones presented here.

## 5. Technical and Scientific Limitations

Before the design of this electrocardiograph began, exhaustive research was conducted to identify possible analog circuits for use in this new device. Different instrumentation amplifiers and Analog-to-Digital Converters (ADC) were analyzed, and ultimately, two Analog Front-End (AFE) solutions, one from Analog Devices [37] and the other from Texas Instruments [48], were the best options found. Between these two options, the Analog Devices ADAS1000 is the only one that explicitly complies with all the necessary norms (ANSI/AAMI/IEC, specifically IEC60601-2-25 and AAMI EC38) and has the configuration flexibility to be integrated into a diagnostic-grade device.

Although it is not ideal to be tied to a particular chipset, the ADAS1000 became the analog core of the Kenshin Device because one of the project’s objectives is to deliver a device that can be used in clinical practice. Since it is not possible to develop a new machine without identifying its concrete components, defining the specific hardware is of vital importance. However, in order to avoid limiting the applicability of this project, the design of new hardware models to suit other applications is planned for development in the near future; these applications include wireless data acquisition, monitoring, remote diagnostics, and ambulatory applications. Indeed, the same analog core (the ADAS1000), is expected to be used on all of these new models until a better option becomes available in the market. When that occurs, the development of legacy support will need to be initiated.

For the reasons just outlined, it is beyond the scope of this project to develop algorithms for use with other types of hardware that deliver signals with characteristics different from the ADAS1000 for now. To address this gap, all the fabrication files of the electrocardiograph proposed here have been made available for collective use through the project webpage, thus enabling this device to be manufactured at the lowest possible cost.

Regarding the use of signals acquired with other hardware within this project, it is expected that those signals will be ANSI/AAMI compliant. That is, they should have less than 30 µV of system noise levels and a sampling rate fast enough to cover the bandwidth of the ECG signals, as described in Section 2; however, their sampling rate should ideally be the same as the sampling rates provided by the ADAS1000. In the future, the development of sampling-rate matching algorithms and adaptive filters might be considered for use with this project’s contributions; but matching sampling rates and equalizing system noise levels is exactly what this proposal wanted to avoid. If a different system can deliver signals that match the signal characteristics delivered by the Kenshin Device, then all signal processing algorithms developed for this project could be used without any incompatibility issues on that other system too.

In terms of signal processing, this project is not limited to using a particular method or approach and welcomes Machine Learning and other Artificial Intelligence branches of study. The general prerequisite while developing algorithms for the Kenshin Project is to anticipate that integration with other algorithms is needed; this prerequisite brings up another important task that still needs to be performed: defining a list of compatibility requirements for signal characteristics, software integration, and hardware. Creating this list will help clarify the limitations of this project, thereby facilitating collaboration.

## 6. Future Challenges

Now that the required hardware for signal standardization is available, the number of challenges ahead is astonishing. The most immediate challenge is the establishment of a new signal databank. To achieve this, the first step is to develop a software tool for signal management. This tool should include features for recording, visualizing, organizing, sharing, commenting, and even editing signals. It is worth noting that clinical practitioners and researchers in the physiological field often have limited familiarity with non-commercial operating systems, and their programming skills may be limited. Therefore, the creation of user-friendly interfaces on widely used operating systems is recommended.

Once the required software tools have been developed, acquiring signals from healthy subjects can be a good start, as sinus rhythms provide plenty of clinical parameters that are of basic use in clinical practice. In fact, the algorithms for the automatic extraction of these last mentioned parameters are the ones that have been reported to have the highest efficiency in academic publications. Consequently, the next immediate challenge is making these algorithms available for community use, either by requesting the source code from the authors or rewriting them. Healthy signals are the easiest to obtain, but for acquiring signals containing pathologies, collaboration with healthcare institutions will indeed be required.

When enough signals have been gathered, a wide spectrum of approaches can be taken to reach the general goal. Methods that emulate human thinking are suggested, but the project is open to any approach or any method that can add value to the collective research. A roadmap to reach the general goal has been designed, as shown in Figure 10. This is not a conventional approach, as a waveform recognition process has been established at the beginning of the process, right after the signal conditioning algorithms. The intention is for the algorithm to first recognize which waveform it is processing, and then extract all the possible parameters for that specific waveform, as this is what a human expert would do. For example, a human cardiologist, before measuring a QRS-complex time duration or an ST-segment elevation, first recognizes the waveform and then proceeds to extract the possible parameters, since different waveforms have different parameters to be measured. Many methods for recognizing waveforms can be developed, but the desired one would be a single method that can be designed to recognize as many different waveforms as possible in a single run, since this is an important requirement for clinical use.

After the waveform has been recognized, a language for describing ECG signals and their parameters is required. This was also suggested in [10]. It would be easier to find pathologies in a systematic language than by directly exploring the signal streaming. Then, of course, automatic diagnostic algorithms will be needed. Detailing the structure of the expert system shown in Figure 10 is actually a matter for another article. The intention of showing it here is to provide a starting point for the future objectives to be achieved.

## 7. Conclusions

This article presents an unusual approach to solving the automatic interpretation problem in ECG signals. It is unusual because the whole problem has been considered in designing a plan that could result in an ECG expert system, instead of focusing on a single specific objective with no integration procedure foreseen. This approach arises from the need to have an academic environment to facilitate collaboration, and integration of different developed algorithms. The aim is to create a single computer application that can assist actual clinical practitioners in clinical practice. The purpose of this proposal is also to provide a means of evaluating academia’s results in real-world applications, and at the same time make these results available to the community, as this is an opportunity to reciprocate what society invests in research.

Automatic interpretation capabilities are already included in many commercial electrocardiographs. Unfortunately, their evaluation reports [7,8,9,10,11,12] can only be assessed because commercial electrocardiographs are not disclosed to the academic community. With this in mind, it is important to note that the project proposed here sets the basis to advance the stage of automatic interpretation capabilities through an open environment of scientific collaboration.

One might assume that this proposal does not have a scientific objective to be pursued. However, after the thousands of papers already published during more than a hundred years of research history, practically since the invention of the electrocardiograph, there is still a lack of a cohesive approach. Therefore, it is now indeed a scientific task to establish a way of putting together all those developed QRS-detection algorithms, model-based filters, and other methods reported with high efficiency into a usable application for evaluating their performance in real clinical practice. Achieving this through an open collaboration project offers two remarkable benefits. The first one is that the objective of leveraging the automatic interpretation efficiency will be reached faster because of the coordinated contributions. The second benefit is that an open electrocardiography system will be created as the result of the academic efforts, for the sake of patients and clinical practitioners alike.

In this document, the proposed solution to the problem of signals incompatibility is the creation of a new standardized signals databank. This solution involved the development of a low-cost but fully compliant electrocardiograph whose design documentation is already published on the project’s website (https://kenshin-ai.org), where feedback is expected and design improvements are welcome. Undoubtedly, new hardware models will be developed in the near future, but the use of the same analog chipset as in the original design is suggested in order to avoid compatibility problems from appearing again. In fact, the selected chipset fully meets all the requirements to support the current state-of-the art regarding ECG interpretation as described in medical books without any problem. Moreover, the used analog chipset also allows for modifying the system architecture to bigger or smaller systems with the same analog features. For example, it is possible to expand the current system architecture to a 15-lead diagnostic electrocardiograph or to reduce it in order to integrate an ambulatory device. It is important to note that the information written in all of the ECG Interpretation books is the accepted results of years of physiological research. This information is not expected to become useless or to change in the coming future, not even when new technologies appear. Therefore, it can be said that investing human resources in pursuing this collective project’s objective is a solid research investment.

Without a doubt, the PhysioNet Project has been the global reference for the ECG signal processing research for years. The Kenshin Project acknowledges this and is grateful to its creators because their work was the inspiration for this project to be born. Fortunately (or unfortunately), there are many ways to solve the same problem, and sometimes perceived solutions differs from one group to another. This is what makes science a way to express human creativity, where universal collaboration always has been intrinsic.

When it comes to the perception of solving a problem, a test called “The Candle Problem” proposed in [49] can come to mind. This test is used for measuring the influence of functional fixedness in participants solving a problem. Functional fixedness is what makes a group of people have the same perception of a problem and therefore propose similar solutions. For example, a box of matches is often seen as a container, not as a support. Hence, sometimes it is good to leave aside a certain expertise in order to reduce functional fixedness. In other words, sometimes it is good to explore new solutions with a fresher mind. In this sense, the perception of the automatic interpretation solution in the Kenshin Project is to emulate the human thinking, instead of finding solutions to solve specific problems. This idea finds its root from an old, but very delightful paper [50], in which Prof. Dreyfus describes how humans acquire skills and how humans go from the general to the specific when learning new skills. He also notes the importance of two different types of knowledge in expert systems and also exposes some medical analogies. All these concepts were considered while designing this proposal.

## Figures and Tables

**Figure 1 diagnostics-14-00600-f001:**
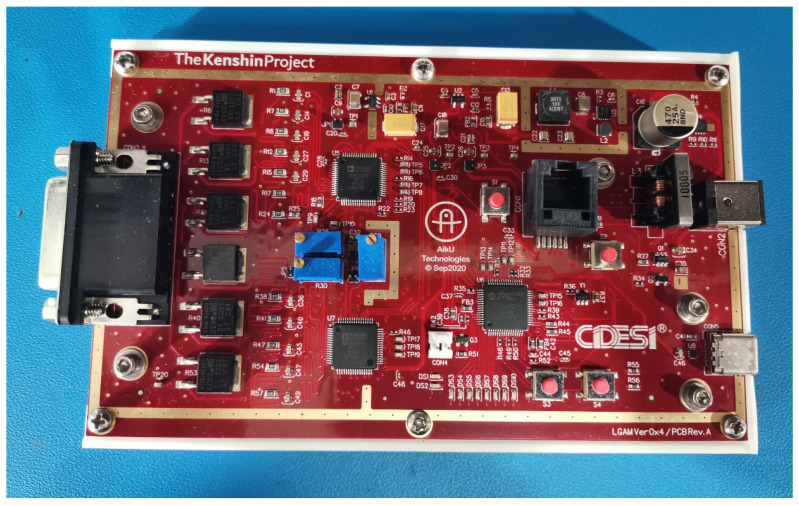
The Kenshin Device PCB (printed circuit board).

**Figure 2 diagnostics-14-00600-f002:**
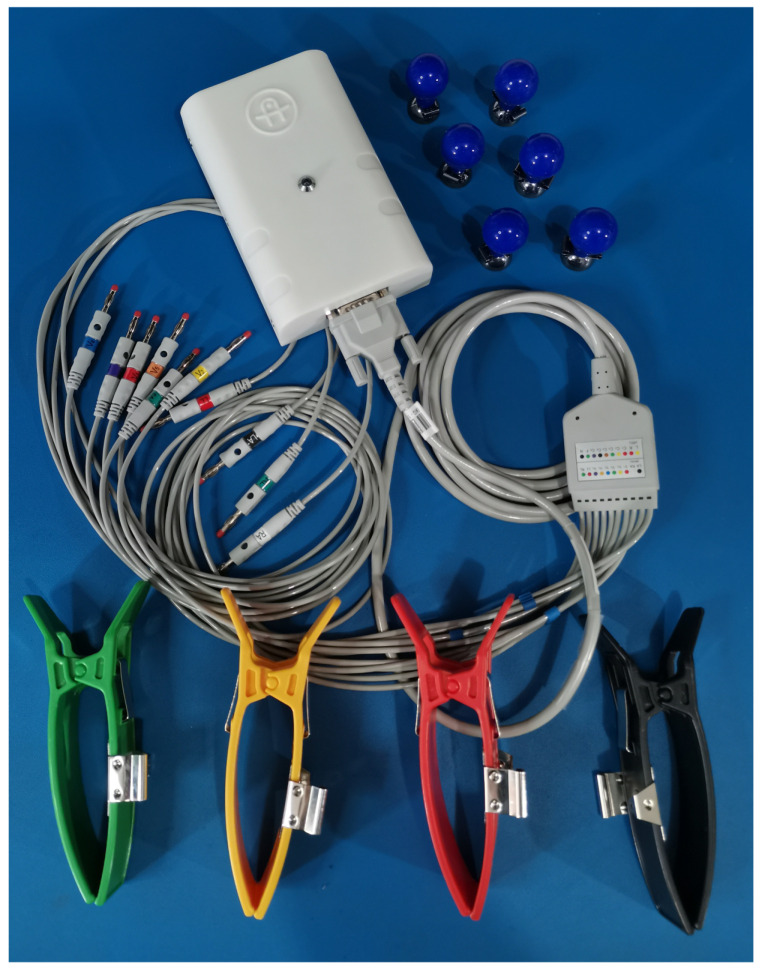
The Kenshin Device (finished product).

**Figure 3 diagnostics-14-00600-f003:**
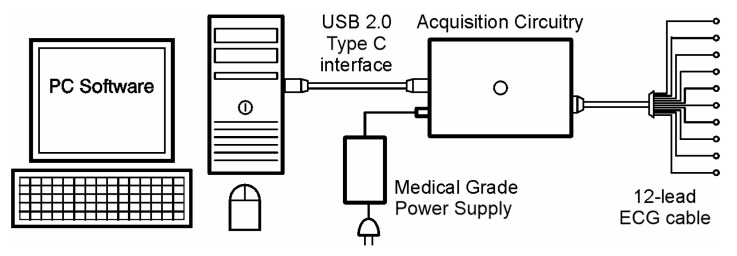
The Kenshin device connectivity diagram.

**Figure 4 diagnostics-14-00600-f004:**
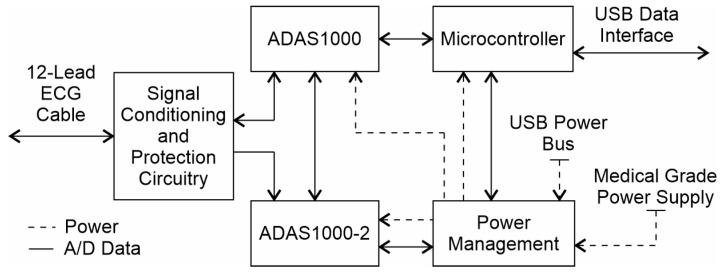
Functional block diagram of the Kenshin Device.

**Figure 5 diagnostics-14-00600-f005:**
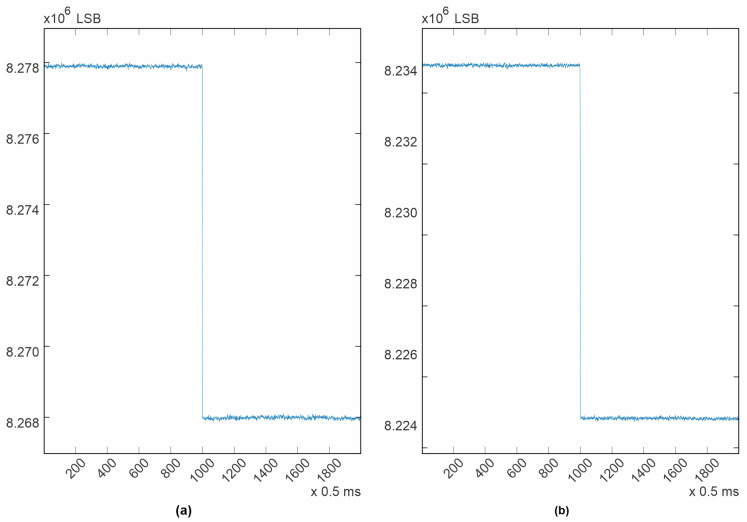
Test tone, 1mV square wave, 1 Hz, one cycle shown, sampling rate 2 kHz. (**a**) LA electrode; (**b**) V3 electrode.

**Figure 6 diagnostics-14-00600-f006:**
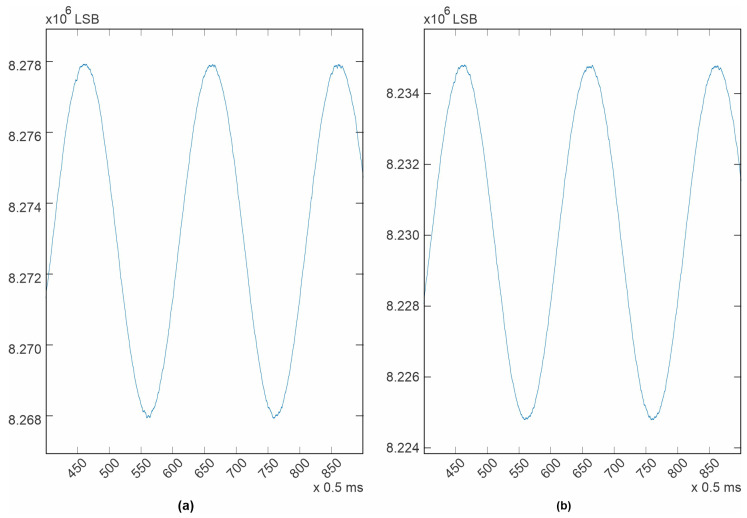
Test tone, 1 mV p-p sine wave, 10 Hz, sampling rate 2 kHz. (**a**) LA electrode; (**b**) V3 electrode.

**Figure 7 diagnostics-14-00600-f007:**
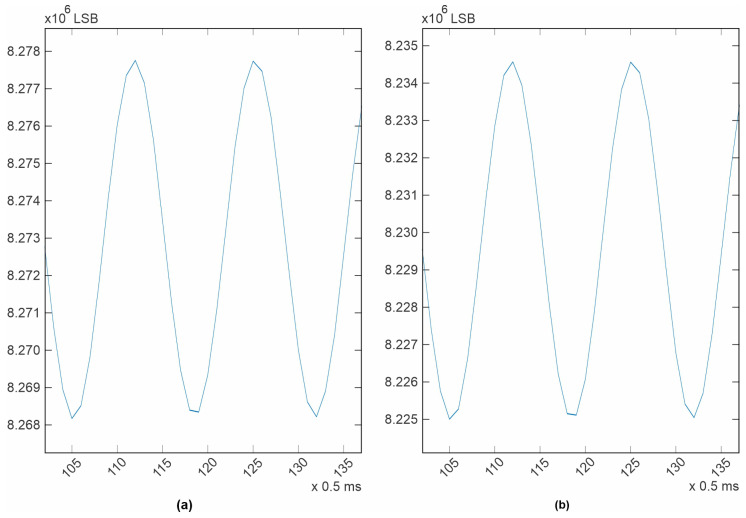
Test tone, 1 mV p-p sine wave, 150 Hz, sampling rate 2 kHz. (**a**) LA electrode; (**b**) V3 electrode.

**Figure 8 diagnostics-14-00600-f008:**
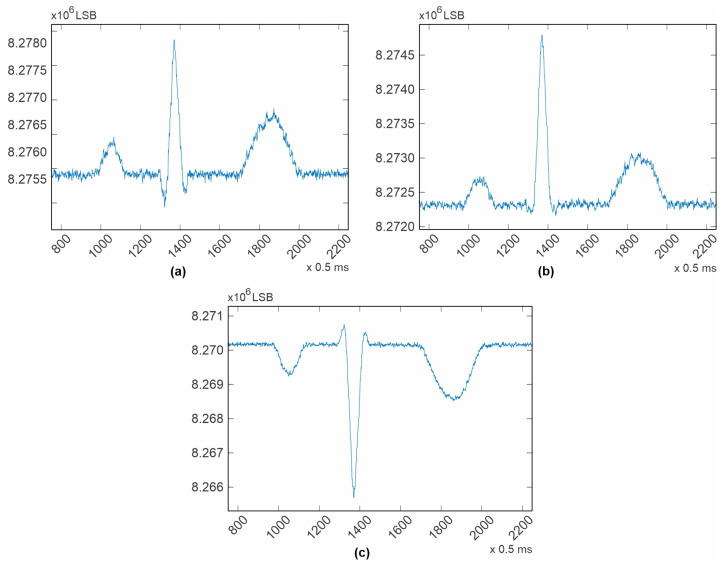
ECG signals from the FLUKE PS420 patient simulator: (**a**) LA electrode, (**b**) LL electrode, (**c**) RA electrode. (**d**) V1 electrode, (**e**) V2 electrode, (**f**) V3 electrode, (**g**) V4 electrode, (**h**) V5 electrode, and (**i**) V6 electrode.

**Figure 9 diagnostics-14-00600-f009:**
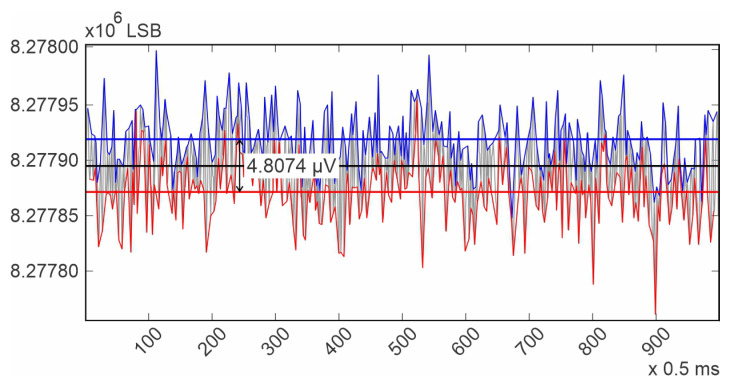
Illustration of the methodology for calculating the system’s noise. In blue, the peaks of the system’s noise and its average value. In red, the valleys of the system’s noise and its average value. In black, the average value of peaks and valleys together.

**Figure 10 diagnostics-14-00600-f010:**
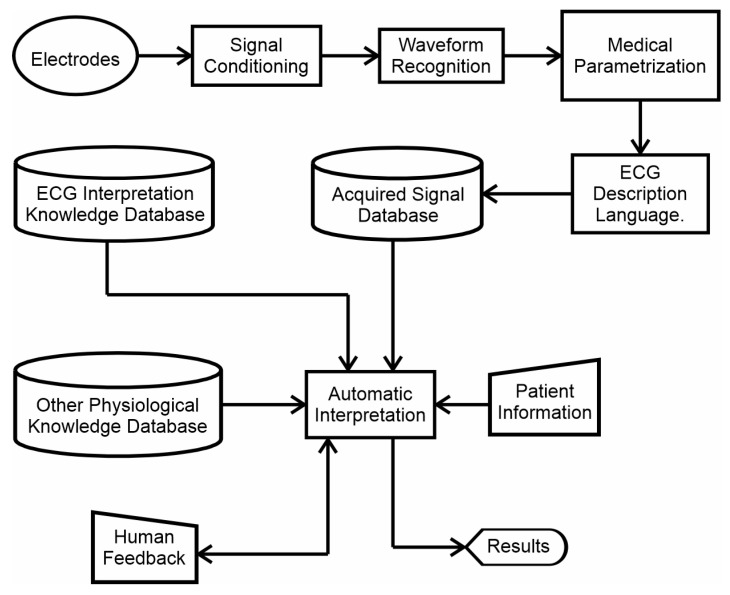
A suggested approach for integrating an electrocardiography expert system.

## Data Availability

The design documentation of the electrocardiograph presented here, along with the signals and MATLAB workspaces, is available at the project webpage (https://kenshin-ai.org).

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
