# Peer review of "A Collaborative Platform for Advancing Automatic Interpretation in ECG Signals"

_diagnostics, 2024, doi:10.3390/diagnostics14060600_

Round 1

Reviewer 1 Report

Comments and Suggestions for Authors

The work is well prepared. Only some minor corrections would make it better.

1-It would be better to give the research gaps in items.

2-Are there any limitations? Future challenges give some clues. But it was challenging for me.

3-There are spelling mistakes in some places. For example, Figure "??" These need to be corrected.

Comments on the Quality of English Language

The quality of english language is good.

Reviewer 2 Report

Comments and Suggestions for Authors

The authors present a new effort to establish a collaborative platform for acquiring, analyzing, and evaluating ECG signals. They present clear and valid motivations for this effort together with the first building blocks of this project. Their open access platform has the potential to overcome current obstacles, i.e., proprietary algorithms and lack of coordination. The proposed project may foster significant progresses in both the academic and clinical communities and I look forward to seeing their progresses in this direction.

The following are a few comments/suggestions to improve the clarity of their manuscript:

1) Although the authors make a compelling argument on why using a standardized hardware is important, this can be seen as a problem too. Forcing the use of a single hardware limits the applicability of this platform and potentially the number of researchers in the community who want to participate. Can the authors:

(a) Comment on this aspect?

(b) Elaborate on plans (if any) to apply their algorithms/platform to signals from other hardware types (maybe by listing a set of compatibility requirements for other hardware types)?

2) An aspect that has not been discussed is the possible role of machine learning algorithms in helping with the interpretation of the ECG signals. Given the increasing role of machine learning models in this field, it would be useful to learn their possible role in this project.

3) Line 386, 432, 433: there are broken links to figures and citations.

4) Figures 5, 6, 7, 8, 9, 10: could you please add the units on all axis in all panels?

5) Figure 6: could you include a more complete and descriptive caption for this figure?

6) It would be better to separate panels (a) through (c) from panels (d) through (f) in Figure 9. All V leads could be presented together in Figure 10 as two figures are anyway necessary to present all the leads.

7) Reference 44 is missing from the reference list. I encourage the authors to carefully read again their work to fix all the small typos, missing citations, and broken links.
